# Factors affecting caregivers' HPV vaccination decisions for adolescent girls: A secondary analysis of a Chinese RCT

Ying Yang[1☉], Yajiao Lu[1☉], Yifan Li[2], Chuanyu Qin[1], Yu He[3], Wenfeng Gong[4], Shenglan Tang[4,5], Dan Wu[2]*, Jing Li[1]*

1 West China School of Public Health and West China Fourth Hospital, Sichuan University, Chengdu, China, 2 Department of Social Medicine and Health Education, School of Public Health of Nanjing Medical University, Nanjing, Jiangsu, China, 3 Yulin Community Health Service Center, Chengdu, China, 4 China Country Office of the Bill and Melinda Gates Foundation, China, 5 Duke Global Health Institute, Duke University, Durham, North Carolina United States of America

☉ These authors contributed equally to this work
* lijingwcph@scu.edu.cn (JL); danwu@njmu.edu.cn (DW)

## Abstract

### Background

Despite the HPV vaccine's effectiveness against cervical cancer, uptake among adolescent girls in China remains low, with caregivers playing a crucial role in vaccination decisions. This study investigates factors influencing caregivers' action to vaccinate their adolescent daughters.

### Methods

Pay-it-forward is a novel model that motivates participants in adopting healthy behaviors and making community contributions. In this study, it offers an individual a free shot of HPV vaccine and then asks whether they would like to donate to support another person to get the same vaccination. This study was embedded in a two-arm randomized controlled trial in China. Potential associated factors were identified based on Anderson's Health Service Utilization Behavioral Model and analyzed through univariate and multivariate binary logistic regression. Caregivers' information, knowledge, attitudes, vaccine confidence, hesitancy, and willingness to vaccinate were collected through online questionnaires. The endpoint was the receipt of the first dose HPV vaccine following an intervention or no intervention (pay-it-forward vs. standard-of-care), which was obtained from an electronic vaccination registry system.

### Results

Among 321 caregivers, 25.9% of their daughters received the HPV vaccine, with 34.2% in the pay-it-forward group and 17.5% in the standard-of-care group.

**Data availability statement:** The date set is available from the figshare database (10.6084/m9.figshare.28424924).

**Funding:** The work reported in this publication is part of the research "Innovation Lab of Work Vaccine Delivery Research", supported by the Bill & Melinda Gates Foundation (INV-034554 to JL). The conclusions and opinions expressed in this work are those of the authors and shall not be attributed to the Foundation. Under the grant conditions of the Foundation, a Creative Commons Attribution 4.0 License has already been assigned to the Author Accepted Manuscript version that might arise from this submission.

**Competing interests:** The authors have declared that no competing interests exist.

Daughters of caregivers who were previously unaware of the HPV vaccine were three times more likely to be vaccinated (*OR*=3.01,95%*CI*:1.27–7.14). Caregivers who did not intend to delay vaccination had daughters with eight times higher vaccination uptake (*OR*=8.26,95%*CI*:4.36–15.67). Participation in the "pay-it-forward" intervention increased vaccination rates by more than twofold (*OR*=2.22,95%*CI*:1.19–4.15). Daughters of unemployed or retired caregivers had nearly four times higher vaccination rates compared to those whose caregivers were employed (*OR*=3.97,95%*CI*:1.81–8.72). Prior refusal of the vaccine by caregivers was associated with an 80% reduction in vaccination uptake among daughters (*OR*=0.23,95%*CI*:0.06–0.81).

## Conclusion

The pay-it-forward intervention, caregivers' knowledge, intention to delay vaccination, occupation, and prior vaccine refusal significantly influence HPV vaccine uptake among adolescent girls in China. Tailored health education, financial support, and community involvement are essential to encourage HPV vaccination among adolescent girls in China.

---

## Introduction

China faces a substantial burden of cervical cancer [1,2]. In 2020, there were over 109,741 new cases and 59,060 deaths from cervical cancer in China, accounting for 18.2% of new global cases and 17.3% of global deaths [1,3]. Notably, incidence rates are increasing among younger populations, likely due to earlier sexual activity and higher-risk behaviors [2, 4]. HPV vaccination has emerged as a highly effective preventive strategy, particularly when administered before sexual debut [4–6]. Despite this, vaccination rates among females under 20 remain alarmingly low, with only 4.00% of 9–14 year-olds and 4.66% of 15–19 year-olds having received their first dose by the end of 2022 [7]. This inadequate coverage presents a considerable obstacle to cervical cancer prevention in China.

Currently, HPV vaccines are not part of China's Expanded Program on Immunization (EPI), which requires caregivers to make out-of-pocket payments [8]. Thus, vaccination decisions predominantly rest with caregivers [9,10]. Existing studies have identified various factors affecting caregivers' willingness to vaccinate their children, including education level, child's age, household income, perceived susceptibility, severity of illness, vaccine confidence and price[11–15]. However, few studies have specifically addressed the factors that influence caregivers' actual action to vaccinate their adolescent daughters. Although a reported 61.0% of parents express willingness to vaccinate their children, actual HPV vaccine uptake among adolescent girls aged ≤18 is only 12.36% [9]. This discrepancy suggests that willingness does not necessarily equate to action, and the determinants of willingness may differ from those influencing actual vaccination behavior. Recent studies from other regions, such as

India, South East Asian and Europe have highlighted similar gaps between vaccine willingness and actual uptake [16–18]. These findings underscore the importance of understanding not only the factors that influence willingness to vaccinate, but also the broader socio-cultural, economic, and healthcare system factors that play a critical role in shaping actual vaccine uptake. Therefore, it is essential to delve into the factors impacting caregivers' real action to vaccinate adolescent girls against HPV. This understanding could enhance HPV vaccine uptake in China and similar public health contexts.

This study addresses this gap by focusing on the actual action of caregivers, thus bridging the divide between willingness and action. In addition, the study introduces a pay-it-forward intervention as a novel approach to influence vaccination behavior, which has not been extensively studied in the context of HPV vaccination. While various interventions, such as health education and financial incentives, have been explored to increase HPV vaccination uptake, they have often faced limitations in overcoming socio-cultural barriers and promoting sustained behavior change [19,20]. In contrast, the "pay-it-forward" model not only provides a tangible benefit (e.g., a subsidized HPV vaccine) but also leverages social reciprocity by encouraging individuals to donate or support others. This unique mechanism taps into intrinsic social motivations, which has shown effectiveness in improving engagement in other health behaviors, such as increasing STI screening rates among men who have sex with men and flu vaccination rates among children and elderly populations [21–23]. The novelty of this intervention lies in its potential to bridge the gap between willingness and actual vaccine uptake by aligning social norms and altruistic behavior with public health goals. This study aims to investigate factors influencing caregivers' action to vaccinate their adolescent girls, based on a pay-it-forward intervention program designed to improve HPV vaccine uptake among adolescent girls in China. By understanding these factors, we aim to inform the development of targeted interventions, guide policymakers and healthcare providers, and support personalized health education programs tailored to diverse population needs. Ultimately, this approach seeks to improve HPV vaccination uptake among adolescent girls in China, thereby contributing to cervical cancer prevention.

## Methods

### Design and participants

The study is embedded in a two-arm randomized controlled trial (RCT) that evaluated the effectiveness of a pay-it-forward intervention on HPV vaccine uptake among adolescent girls in Chengdu, China. For the purposes of this study, the term "caregivers" is used to refer to adults who are responsible for making healthcare decisions for adolescents, including but not limited to biological parents. This broader term encompasses individuals such as guardians, grandparents, or other family members involved in caregiving roles. The detailed methodology and design have been previously described in detail in the protocol publication [24]. The RCT was conducted at four community health service centers in Chengdu, China between July 6, 2022 and June 9, 2023. The four sites were selected to reflect varying socio-economic contexts based on the relative level of individual disposable income in 2021. This approach ensures a broad representation of different socio-economic environments within Chengdu. The selected sites are as follows: Site A: Located in the most developed urban area, this site benefits from higher healthcare access and represents a higher socio-economic status with better public health infrastructure. Site B: Situated in a higher middle-income suburban area, this site experiences relatively good healthcare access but is influenced by suburban socio-economic factors. Site C: This site represents a lower middle-income suburban area with more limited healthcare resources and socio-economic challenges. Site D: The least developed area in terms of infrastructure, healthcare access, and socio-economic conditions. This site faces the greatest challenges related to economic factors and healthcare delivery. Eligibility criteria for participation included (1) girls aged 15–18 years; (2) residents of the communities served by the selected health centers; (3) no prior history of HPV vaccination; (4) no history of vaccine allergy; and (5) caregivers' consent to participate in the study. Girls who were under 15 or over 18 years of age, as well as those who did not consent to complete the questionnaire, were excluded.

 

We obtained a de-identified contact list of adolescent girls aged 15–18 years from the residential committee through community health center staff. This list included all girls within this age range who have registered their names and contacts information at the community health center. We then performed simple random sampling to select eligible girls by generating a random number for each individual using Microsoft Excel. These individuals were sorted in ascending order based on their assigned random numbers. Recruitment proceeded by contacting individuals at the top of the list until the desired sample size was reached at each study site.

Upon arrival at the study site, participants were randomly assigned to either the standard-of-care arm or the pay-it-forward arm using sealed opaque envelopes to ensure allocation concealment. Each envelope contained a random numerical ID corresponding to the assigned arm (standard-of-care or pay-it-forward). The allocated intervention was concealed from both study participants and research assistants onsite prior to assignment until the envelope was opened, thereby maintaining blinding throughout the assignment process.

During the intervention process, all participants received an educational pamphlet about cervical cancer and HPV vaccines. In addition, participants in the pay-it-forward arm received personalized postcard messages containing additional information about HPV vaccines and well-wishes from others, designed based on caregiver preferences gathered during focus group discussions. Caregivers who chose to vaccinate their daughters received a subsidy of CNY 330(approximately USD 46.21) towards the HPV vaccine. The total cost for a full three-dose HPV vaccine series in the local market ranged from approximately CNY 990 to CNY 3,900 (USD 139–546), depending on the type and manufacturer. After vaccination, they had the opportunity to donate and/or write postcard messages to encourage other to get vaccinated early. Participants in the standard-of-care arm who chose to receive the HPV vaccine had to pay the standard market price for the HPV vaccine without any subsidies or additional messaging. All participants completed an online questionnaire at study site (Appendix A1) after the intervention. The sample size was estimated based on a pilot study conducted in Chengdu from January to February 2022. In this pilot study, vaccine uptake was observed to be 98% in the pay-it-forward arm and 82% in the standard-of-care arm, with a 16% difference. However, due to the low HPV vaccination rate (<10%) in the target population, we conservatively estimate 20% uptake in the standard-of-care and 36% in the pay-it-forward arm for the trial. To detect this difference with 80% power, 120 participants are required per arm. Considering a 10% non-response or dropout rate based on results from the pilot, 133 participants per arm are needed. To allow for secondary and subgroup analyses, the desired sample size will be increased by 20%, resulting in a total of 320 participants. A total of 321 participants were enrolled in the study, with 160 in the standard-of-care arm and 161 in the pay-it-forward arm.

## Data sources

Data for this study were derived from the randomized controlled trial, including participant questionnaires and vaccination records maintained by the study staff. The primary data source was a structured online questionnaire completed by caregivers after providing informed consent. The questionnaire encompassed sections on sociodemographic information, willingness to vaccinate within a month, vaccine confidence, vaccine hesitancy, and intention to delay vaccination if 4v- and/or 9v- vaccines were unavailable. Adolescents' vaccination records were retrieved from an electronic vaccination registry system within three months after the pay-it-forward intervention. Only participants with complete questionnaire data were included.

## Variable measurements

**Guiding Conceptual framework.** The study was guided by Anderson's Health Service Utilization Behavioral Model, which has been widely used to understand factors influencing health service utilization, including sexual health services [25,26]. The model posits that health service use is determined by a combination of predisposing, enabling, need, and environmental factors [27]. Predisposing factors include demographic characteristics, social structure variables, beliefs, attitudes, and knowledge about health services [28]. Enabling factors include financial resources, access to services, and

availability of healthcare facilities [28]. Need factors include perceived or evaluated health conditions requiring medical attention [28]. Environmental factors include external influences such as social networks and exposure to information [28]. An adapted version of the model was used to select independent variables and interpret the results (Fig 1).

**Outcome variable.** The primary outcome was the HPV vaccination behavior of the adolescents, measured by the receipt of the first dose of the HPV vaccine within a three-month period following the intervention. Due to the unavailability of the 4v-HPV and 9v-HPV vaccines during the study period, all adolescents received the 2v-HPV vaccine.

**Independent variables.** Based on the adapted Andersen's model, the following independent variables were assessed: predisposing factors including caregivers' sociodemographic information (i.e., age, gender, education and marital status); Attitude towards the HPV vaccine (i.e., perceived importance, safety, effectiveness and intention to delay vaccination); Awareness of the HPV vaccine (whether the caregiver had heard of HPV and the HPV vaccine) and its price. Caregivers' intention to delay vaccination was assessed using a scenario-based question about vaccine preference among the available HPV vaccines (2v-,4v-, and 9v-). To ensure that this measure accurately captures the concept of intention to delay vaccination, we conducted a pilot study in which the wording of this question was pre-tested for clarity and understanding with a small group of caregivers from the study's target population. In the context of China's limited supply and difficulty in securing appointments for 4v- and 9v-HPV vaccines, caregivers who preferred the 2v- vaccine or were willing to accept alternatives if their preferred vaccine was unavailable were classified as not intending to delay vaccination. Those who chose to wait exclusively for the 4v- or 9v- vaccines were considered as intending to delay vaccination. Enabling factors include the annual household income, the occupation of the caregiver and participation in the pay-it-forward intervention. Need factors included a family disease history (Whether a family member had been infected with HPV or diagnosed with cervical cancer) and vaccine hesitancy. Vaccine hesitancy was assessed by asking participants to indicate their level of agreement with three statements regarding past behavior and attitudes: "Have you ever been hesitant about getting the HPV vaccination (except for allergies)?"; "Have you ever postponed the HPV vaccination (except for allergies)?"; and "Have you ever refused to receive the HPV vaccination (except for allergies)?"

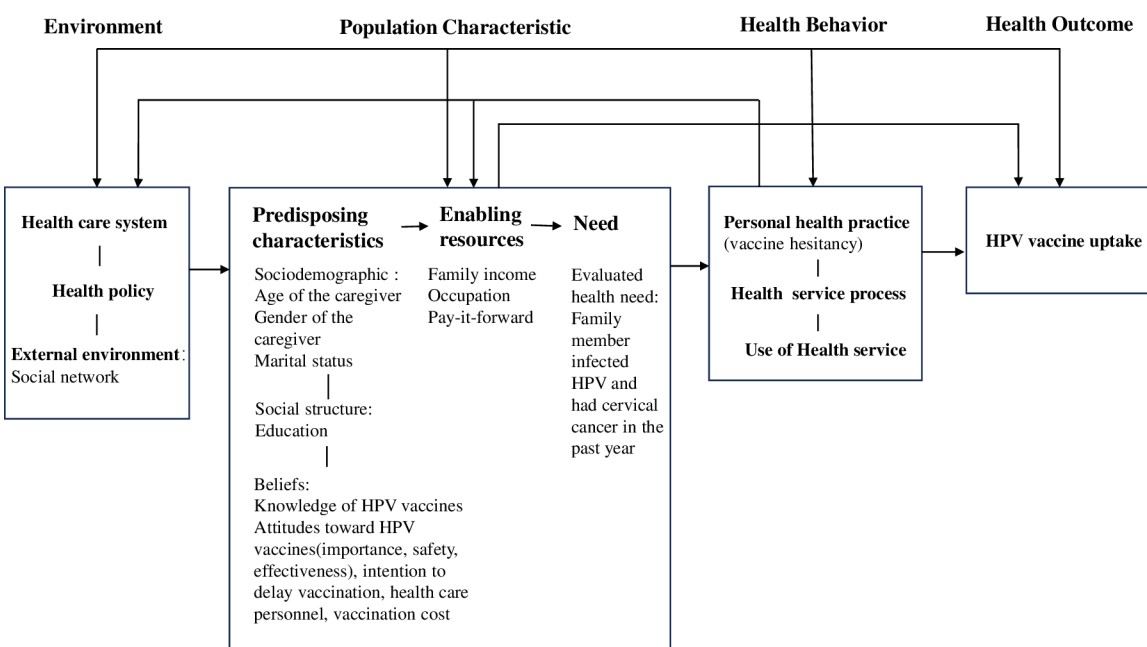

**Fig 1. Adapted Andersen's Health Service Utilization Behavioral Model of potential factors associated with HPV vaccine uptake.**

Each response recorded separately to reflect different aspects of past vaccine-related decision-making."External environmental factors focused on social influences, including HPV vaccination status of friends or relatives, exposure to negative information about the HPV vaccine, attitudes of relatives and friends toward the HPV vaccine, and any adverse experiences after HPV vaccination within their social circle.

## Statistical analysis

Descriptive statistics, including means and frequencies, were calculated for demographic variables and HPV vaccine uptake. Univariable logistic regression analyses were performed to assess the association between each independent variable and HPV vaccine uptake, expressing the results as odds ratios (*OR*s) with 95% confidence intervals (*CI*s). Variables with a *p*-value < 0.20 in the univariable analyses were considered for inclusion in the multivariable logistic regression model. The multivariable regression model was developed using a backward stepwise strategy, adjusting for potential confounders such as age and gender. This method was chosen for its systematic elimination of non-significant variables and ability to simplify the model. However, it carries risks of overfitting, exclusion of important interactions, and dependence on the initial variable set. The logistic regression models in this study assumed the independence of observations, linearity of the logit, and the absence of multicollinearity. To ensure the independence of observations, participants were randomized at the individual level, and no repeated measures were taken from the same individual. Multicollinearity was assessed using the variance inflation factor (VIF), and variables with a VIF > 5 were excluded. A *p*-value < 0.05 was considered statistically significant. All analyses were conducted using IBM SPSS Statistics version 26.0.

## Ethical approval

The Ethics Committee of West China Fourth Hospital and West China School of Public Health approved the study (Gwll2021057/ Gwll2023125). Informed written consent was obtained from all participants before participation. This study was registered on the Chinese Clinical Trial Registry (ChiCTR2200055542).

## Results

### Caregiver characteristics and HPV vaccination status of adolescents

All 321 caregivers were sent the questionnaire, with a 100% completion rate in both the pay-it-forward arm and the standard-of-care arm. Therefore, all 321 caregivers were included in the final analysis. The mean age of caregivers was 44.7 years (*SD* = 8.6). The majority of caregivers (80.1%, 157/321) were female. Regarding occupation, 43.3% of caregivers were employed, and 32.1% were farmers. Most (91.9%) caregivers had education level above primary school and the majority (80.4%) of caregivers were married. Of all participating in the study, 25.9% (82/321) of adolescent girls received their first HPV vaccine dose within three months, with 17.5% in the standard-of-care arm and 34.2% in the pay-it-forward arm. (Table 1)

### Factors influencing caregivers' action to vaccinate their daughters

The univariable logistic regression results are shown in the Table 2. Caregivers who had previously heard of HPV or the HPV vaccine were more likely to vaccinate their daughters. Caregivers with intention to delay vaccination also had higher vaccination rates for their daughters. However, caregivers with an annual household income exceeding USD 1,120 were less likely to have their adolescents vaccinated. Adolescents whose caregivers were farmers, unemployed, or retired had higher vaccination rates compared to those whose caregivers were employed. Additionally, adolescents who received the pay-it-forward intervention had a higher HPV vaccine uptake than those did not received any intervention. Conversely, girls whose peers had been vaccinated against HPV were less likely to be vaccinated.

**Table 1. Caregiver characteristics and HPV vaccination status of adolescents (N = 321).**

| Characteristics | Total n (%*) | Unvaccinated n (%†) | Vaccinated n (%†) |
|---|---|---|---|
| Age (Years)‡ | 44.7 ± 8.6 | 45.1 ± 8.7 | 43.7 ± 8.3 |
| Gender | | | |
| Female | 257(80.1) | 185(72.0) | 72(28.0) |
| Male | 64(19.9) | 53(82.2) | 11(17.2) |
| Occupation | | | |
| Employed | 139(43.3) | 116(83.5) | 23(16.5) |
| Farmer | 103(32.1) | 73(70.9) | 30(29.1) |
| Unemployed or retired | 79(24.6) | 49(62.0) | 30(38.0) |
| Education of the caregiver | | | |
| Primary school | 26(8.1) | 17(65.4) | 9(34.6) |
| Middle school | 205(63.9) | 147(71.7) | 58(28.3) |
| Undergraduate or above | 90(28.0) | 74(82.2) | 16(17.8) |
| Family income levels | | | |
| >1120 USD/year | 88(27.4) | 74(84.1) | 14(15.9) |
| <=1120 USD/year | 233(72.6) | 164(70.4) | 69(29.6) |
| Marital status | | | |
| Unmarried | 37(11.5) | 29(78.4) | 8(21.6) |
| Married | 258(80.4) | 191(74.0) | 67(26.0) |
| Others | 26(8.1) | 18(69.2) | 8(30.8) |
| Pay-it-forward | | | |
| No | 160(49.8) | 132(82.5) | 28(17.5) |
| Yes | 161(50.2) | 106(65.8) | 55(34.2) |

*The percentage of the total study population

†The percentage of vaccinated or unvaccinated individuals within different characteristic groups

there were 19 missing values for Variable age, Y missing values for Variable 2, and Z missing values for Variable 3

‡ There were 19 missing values for the variable "age"

Variables with a *P*-value less than 0.2 after adjusting for age and gender of caregivers were included in the multivariable model. The multivariable logistic regression analyses revealed that knowledge of HPV vaccine, intention to delay vaccination, caregiver occupation, the pay-it-forward and prior refusal of the vaccine remained statistically significantly associated with HPV vaccine uptake. Caregivers who had not heard of the HPV vaccine prior to participating in our program were more likely to vaccinate their daughters than those who were already aware of it (*OR*=3.01,95%*CI*:1.27–7.14, *P*=0.013). Adolescents whose caregivers did not intend to delay vaccination were nearly eight times higher vaccine uptake than those whose caregivers did (*OR*=8.26,95%*CI*:4.36–15.67, *P*<0.001). Adolescents who received the pay-it-forward intervention had twice the HPV vaccine uptake compared to those who did not (*OR*=2.22,95%*CI*:1.19–4.15, *P*=0.013). Furthermore, adolescents whose caregivers who were unemployed or retired had higher vaccination rates than those whose caregivers were employed (*OR*=3.97,95%*CI*:1.81–8.72, *P*=0.001). Lastly, adolescents who had previously refused the HPV vaccine for some reason (other than allergy or illness) had lower vaccine uptake (*OR*=0.23,95%*CI*:0.64–0.81, *P*=0.022). (Table 2)

## Discussion

Administering the HPV vaccine during adolescence maximizes its efficacy in preventing cervical cancer [4,29]. This highlights the pivotal role of caregivers in facilitating vaccination decisions for their daughters [4,9,29,30]. Unlike prior studies

**Table 2. Associated factors for caregivers' action to vaccinate adolescent girls against HPV (N = 321).**

| Explanatory variables ‡ | Univariable analysis | | Multivariable analysis | | Direction of Association |
|---|---|---|---|---|---|
| | OR (95% CI) | P | OR (95% CI) | P | |
| *Predisposing factors* | | | | | |
| **Age of caregivers** | 0.98(0.95-1.01) | 0.167 | | | |
| **Gender of caregivers** | | | | | |
| Female | Ref. | 0.080 | | | |
| Male | 0.53(0.26-1.08) | | | | |
| **Education** | | | | | |
| Primary school | Ref. | 0.098 | | | |
| Middle school | 0.75 (0.31-1.77) | | | | |
| Undergraduate or above | 0.41 (0.16 −1.08) | | | | |
| **Have you ever heard of HPV** | | | | | |
| Yes | **Ref.** | **0.012** | | | |
| No | **2.54(1.23-5.27)** | | | | |
| **Have you ever heard of the HPV vaccine** | | | | | |
| Yes | **Ref.** | **0.002** | **Ref.** | **0.013** | |
| No | **3.15(1.54-6.45)** | | **3.01(1.27-7.14)** | | Positive |
| **Intention to delay vaccination** | | | | | |
| Yes | **Ref.** | **<0.001** | **Ref.** | **<0.001** | |
| No | **7.54(4.28-13.27)** | | **8.26(4.36-15.67)** | | Positive |
| **Is price a barrier** | | | | | |
| No | Ref. | 0.060 | Ref. | 0.082 | |
| Yes | 1.65(0.98-2.78) | | 1.78(0.93-3.41) | | |
| *Enabling factors* | | | | | |
| **Annual family income** | | | | | |
| >1120 USD/year | **Ref.** | **0.014** | | | |
| <=1120 USD/year | **0.45(0.24-0.85)** | | | | |
| **Occupation** | | | | | |
| Employment | **Ref.** | **0.002** | **Ref.** | **0.003** | |
| Farmer | **3.09(1.63-5.84)** | | **3.97(1.81-8.72)** | **0.001** | Positive |
| Unemployed/retired | **2.07(1.12-3.84)** | | 1.88(0.89-3.94) | 0.097 | |
| **Pay-it-forward** | | | | | |
| No | **Ref.** | **0.001** | **Ref.** | **0.013** | |
| Yes | **2.45(1.45-4.12)** | | **2.22(1.19-4.15)** | | Positive |
| *Need factors* | | | | | |
| **Has HPV vaccination been delayed for some reason (other than allergy or illness)** | | | | | |
| No | Ref. | 0.569 | | | |
| Yes | 84(0.47-1.52) | | | | |
| **Whether HPV vaccination was refused for some reason (other than allergy or illness)** | | | | | |
| No | Ref. | 0.387 | **Ref.** | **0.022** | |
| Yes | 0.66(0.26-1.68) | | **0.23(0.06-0.81)** | | Negative |
| *External environment factors* | | | | | |
| **Is there anyone around who has been vaccinated against HPV** | | | | | |
| No | **Ref.** | **0.033** | | | |
| Yes | **0.53(0.29-0.95)** | | | | |

*(Continued)*

**Table 2.** (Continued)

| Explanatory variables ‡ | Univariable analysis | | Multivariable analysis | | Direction of Association |
|---|---|---|---|---|---|
| | OR (95% CI) | P | OR (95% CI) | P | |
| Adverse reactions to HPV vaccination of friends or relatives | | | | | |
| No | Ref. | 0.159 | | | |
| Yes | 0.34(0.08-1.52) | | | | |

‡ The table presents only the explanatory variables that were included in the multivariable model, specifically those with a *P*-value less than 0.2 after adjusting for age and gender of caregivers. Statistically significant variables of the univariate analysis are also included.

focusing on caregivers' willingness, our research specifically examined actual vaccination behaviors to identify determinants influencing HPV vaccine uptake among adolescent girls. We found that engagement in the pay-it-forward intervention, caregiver occupation, knowledge of the HPV vaccine, intentions to delay vaccination, and prior vaccine refusal were significantly associated with vaccination uptake. These insights can inform the development of targeted strategies to enhance HPV vaccination rates.

## Impact of the pay-it-forward intervention

Our findings revealed that caregivers participating in the pay-it-forward intervention had a substantially higher vaccination rate (34.2%) compared to those receiving standard care (17.5%). This suggests that the pay-it-forward model is an effective approach to increase HPV vaccine uptake, corroborating previous studies demonstrating the efficacy of such models in improving health service utilization [22,23,31]. The effect of the pay-it-forward intervention was quantified by an odds ratio (*OR*) of 2.22 (95% *CI*: 1.19–4.15), meaning that caregivers in the pay-it-forward group were 2.22 times more likely to vaccinate their daughters compared to those in the standard-of-care group. While this effect is significant, it is less pronounced than some other factors, such as the intention to delay vaccination (*OR* = 8.26), which emerges as a key barrier to vaccination. Several mechanisms may explain this success. First, the intervention provided tailored educational materials, such as specially designed postcards. These materials may have addressed specific knowledge gaps among caregivers, enabling more informed decision-making. Second, the financial subsidies inherent in the pay-it-forward model likely reduced economic barriers. Prior research shown that lowering vaccine costs increases parental willingness to vaccinate [32]. Third, the altruistic component of the model may have fostered a sense of community engagement and trust, positively influencing health behaviors [33,34]. Interestingly, the standard-of-care arm exhibited a higher uptake (17.5%) than previous observational studies (4.66%) [7]. This highlights the impact of active engagement strategies.

These results underscore the potential of the pay-it-forward intervention to expand HPV vaccine coverage, particularly among populations not currently served by government-funded programs. While some regions in China have initiated free or subsidized HPV vaccination initiatives[35], many adolescents remain unvaccinated due to geographic and socioeconomic disparities. Implementing the pay-it-forward model could bridge this gap by combining financial assistance with community engagement. However, due to the strong intention to delay vaccination (*OR* = 8.26), future research should dissect the individual components of the intervention and focus on strategies to address this barrier in particular. Additionally, our observation that unemployed or retired caregivers were nearly 4 times more likely to vaccinate their daughters compared to employed caregivers, and that those previously unaware of the HPV vaccine were 3 times more likely to do so. These results may indicate a higher level of trust in the healthcare system among these population and the importance of informational campaigns.

## Barriers to HPV vaccine uptake

Caregivers who intended to delay HPV vaccination were more than 8 times as likely to postpone vaccination compared to those who did not intend to delay. This factor represents a significant barrier in the Chinese context and should be prioritized in interventions. Despite a high willingness to vaccinate within a month among caregivers (98.8%) (317/321) (S1 Fig), actual vaccine uptake was markedly lower (25.9%) (83/321). This highlights a significant gap between willingness and action. This disparity may be attributed to vaccine availability issues, particularly the preference for 4v- and 9v- HPV vaccines [36,37]. Although the domestically produced 2v-HPV vaccines were available, many caregivers opted to wait for the higher-valency vaccines. These vaccines often face supply constraints when our study was conducted. The demand for higher-valency vaccines, coupled with limited availability, may lead to vaccination delays or refusals [38,39]. Furthermore, caregivers who had previously refused the HPV vaccine were less likely to vaccinate their daughters, possibly due to persistent misconceptions or lack of confidence in the vaccine's safety and efficacy [40]. This finding is particularly concerning because those with prior refusals ($OR$ = 0.23) are less likely to change their position. Addressing vaccine hesitancy and building confidence in vaccine safety may require more intensive efforts. Misinformation propagated through social media regarding vaccine cost and availability may exacerbate these concerns [37]. Both 2v- and 9v-HPV vaccines offer protection against HPV types 16 and 18. These types are responsible for the majority of cervical cancer cases [41,42]. Therefore, it is imperative to emphasize the benefits of early vaccination with any available vaccine. Enhancing public education on vaccine efficacy and addressing supply chain issues for higher-valency vaccines are critical steps toward improving uptake. In addition to misinformation and vaccine availability, cultural stigma significantly impacts HPV vaccine uptake [43,44]. In China, cultural concerns about adolescent sexuality and the HPV vaccine's association with sexual behavior contribute to vaccine hesitancy. Similar stigma is seen in other Asian countries, like Hong Kong and South Korea, though its intensity varies. In India, seeking the HPV vaccine can be stigmatized due to concerns over family honor and social prestige, further fueling vaccine hesitancy [44]. This stigma can deter caregivers from vaccinating adolescents due to concerns about societal judgment or misinterpretation of their intentions [44]. To address this, public health campaigns should emphasize the vaccine's role in preventing cervical cancer while incorporating culturally sensitive messaging while incorporating culturally sensitive messaging. These efforts can help to reduce stigma, promote acceptance, and improve vaccination uptake.

## Global relevance and application of the findings

HPV vaccination rates remain low in many countries, particularly in low-income regions where vaccine access is limited and healthcare systems may lack the infrastructure to promote the vaccine effectively [45]. The significant role of economic barriers and lack of knowledge about the vaccine mirrors challenges observed in other countries, such as India and several sub-Saharan African nations [46,47]. Although this study was conducted in China, the findings are highly relevant to other regions facing similar barriers to HPV vaccination uptake. Countries in Sub-Saharan Africa, Southeast Asia, and rural areas of Latin America, where financial and informational barriers to vaccination are prevalent, could benefit from strategies that combine financial support with community-driven health education initiatives. For example, in Southeast Asia, where cultural stigmas around adolescent sexuality are prevalent, the "pay-it-forward" model could be adapted by involving local leaders, such as religious figures or community health workers, to help reshape societal perceptions around the HPV vaccine. Future research could explore how the "pay-it-forward" model might be adapted to improve HPV vaccination rates in diverse settings. This would involve tailoring the intervention to suit local contexts, cultural nuances, and available resources.

## Limitations

Our study has several limitations. As a study embedded within a randomized controlled trial, the sample size was relatively small and limited to an urban sample, which may restrict the generalizability of the findings to rural areas or other regions with different socio-cultural contexts. Additionally, the intervention at one site coincided with the COVID-19

pandemic, which may have reduced participants' willingness to visit clinics due to restricted access and increased anxiety about public spaces. The new guidelines imposed on healthcare settings may have led to longer waiting times, restricted clinic hours, and limited patient capacity, which further dissuaded caregivers from seeking vaccination services for their children. These factors likely contributed to lower vaccination rates at Site D compared to other sites. Another limitation is the self-reported nature of the data, which may be affected by recall bias and social desirability bias, as caregivers might not accurately remember past vaccination events or may report more positive vaccine confidence to align with social expectations. Frequent stockouts of the 4v- and 9v- HPV vaccines may have also led to underestimation of vaccine uptake due to this structural barrier. Nonetheless, alternative HPV vaccine options were available across all sites, ensuring access to vaccination for participants considering other formulations.

## Conclusions

Our study identifies the following key factors influencing HPV vaccine uptake among adolescents:

1.  Effectiveness of the pay-it-forward intervention – Higher vaccination rates were observed in the pay-it-forward arm compared to the standard-of-care arm.

2. Caregivers' knowledge of HPV vaccines – Understanding vaccine safety and effectiveness is crucial for promoting vaccination.

3. Intentions to delay vaccination – Caregivers who intended to delay vaccination were less likely to vaccinate their daughters.

4. Caregiver occupation – Caregivers' employment status influenced their decision to vaccinate.

5. Prior vaccine refusal – Caregivers who previously refused the vaccine were less likely to vaccinate.

These findings highlight the need for tailored health education to address specific knowledge gaps and misconceptions, especially among caregivers in underserved communities. Programs should focus on dispelling myths about the HPV vaccine, emphasizing its safety and effectiveness, and fostering community engagement through local health workers. These workers should be trained in vaccine knowledge, effective communication, and cultural sensitivity to organize outreach activities that build trust and increase vaccine uptake. Financial subsidies are essential to reduce economic barriers for low-income families. For example, a 45 USD subsidy, based on the cost of a domestic 2v-HPV vaccine, could encourage early initiation of vaccination and be implemented through government programs or local health insurance schemes to ensure equitable access. By leveraging these strategies, future public health efforts can further refine interventions to significantly enhance HPV vaccination rates and contribute to cervical cancer prevention, both in China and in similar public health contexts globally.

However, there are several limitations in this study. The relatively small and urban-focused sample size may restrict the generalizability of our findings to rural or socioeconomically diverse populations. Additionally, potential biases may arise from the socioeconomic distribution of participants, which could influence the observed associations. Future research should aim to address these limitations by conducting larger, more diverse cross-sectional studies to examine how these factors influence vaccination decisions across regions and communities. Mixed-methods studies could explore cultural, socioeconomic, and psychological barriers to HPV vaccination, using caregiver interviews to explore the reasons for vaccination decisions, and delays in vaccination.

## Supporting information

**S1 File. Supplementary materials used in primary study.**
(DOCX)

**S1 Table. Collinearity diagnosis (N= 321).**
(DOCX)

**S1 Fig. Caregivers' willingness to vaccinate girls against HPV and girls' actual vaccination rate.**
(DOCX)

## Acknowledgments

The authors are grateful to all participants and extend their thanks to the collaborative health staff at Yulin Community Health Service Center, Longtan Community Health Service Center, Xinjin District Maternal and Child Healthcare Hospital, and the Third People's Hospital of Chengdu Eastern New Area for their valuable contributions. We thank all advisory group members who provided advice on contextual background.

## Author contributions

**Conceptualization:** Dan Wu, Jing Li.

**Data curation:** Jing Li.

**Funding acquisition:** Wenfeng Gong, Shenglan Tang.

**Investigation:** Ying Yang, Yajiao Lu, Yifan Li, Chuanyu Qin, Dan Wu, Jing Li.

**Methodology:** Ying Yang, Yajiao Lu, Yifan Li, Chuanyu Qin, Yu He, Dan Wu, Jing Li.

**Resources:** Yu He, Wenfeng Gong, Shenglan Tang.

**Writing – original draft:** Ying Yang, Yajiao Lu, Dan Wu, Jing Li.

**Writing – review & editing:** Ying Yang, Yajiao Lu, Yifan Li, Chuanyu Qin, Yu He, Wenfeng Gong, Shenglan Tang, Dan Wu, Jing Li.

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
