## [Decision Letter · Decision Letter 0]

PONE-D-24-51311Factors Influencing Caregivers’ Action to Vaccinate Adolescent Girls Against HPV in ChinaPLOS ONE

Dear Dr. Li,

Thank you for submitting your manuscript to PLOS ONE. After careful consideration, we feel that it has merit but does not fully meet PLOS ONE’s publication criteria as it currently stands. Therefore, we invite you to submit a revised version of the manuscript that addresses the points raised during the review process.

**ACADEMIC EDITOR: **

Dear Authors,

the manuscript needs major revisions, please respond point by point to the reviewers' requests.

Kind regards

We look forward to receiving your revised manuscript.

Kind regards,

Omar Enzo Santangelo

Academic Editor

PLOS ONE

Journal Requirements:

“This project is supported by Bill & Melinda Gates Foundation (INV-034554), which is responsible for providing financial support to help complete this project. All opinions, findings, conclusions and recommendations in this article are those of the authors and do not represent the views of Bill & Melinda Gates Foundation.”

“The Bill & Melinda Gates Foundation (INV-034554, INV-003174) funded this study. DH and WFG from Gates provided vaccine-related expertise and relevant contexts, but did not involve in the study design, implementation, data collection or data analysis.”

“This project is supported by Bill & Melinda Gates Foundation (INV-034554), which is responsible for providing financial support to help complete this project. All opinions, findings, conclusions and recommendations in this article are those of the authors and do not represent the views of Bill & Melinda Gates Foundation.”

Reviewers' comments:

Reviewer's Responses to Questions

**Comments to the Author**

1. Is the manuscript technically sound, and do the data support the conclusions?

Reviewer #1: Yes

Reviewer #2: Yes

2. Has the statistical analysis been performed appropriately and rigorously? 

Reviewer #1: Yes

Reviewer #2: Yes

3. Have the authors made all data underlying the findings in their manuscript fully available?

Reviewer #1: Yes

Reviewer #2: Yes

4. Is the manuscript presented in an intelligible fashion and written in standard English?

Reviewer #1: Yes

Reviewer #2: Yes

5. Review Comments to the Author

Reviewer #1: This study addresses a critical public health issue by investigating factors influencing HPV vaccination uptake among adolescent girls in China, a country facing significant cervical cancer burden. The findings provide valuable insights into caregiver decision-making and the impact of innovative interventions like the "pay-it-forward" model. However, methodological clarity, result interpretation, and presentation need improvement. I recommend a Major Revision.

Abstract

- Clarify the "pay-it-forward" intervention in a single concise sentence for accessibility to a broader audience.

Introduction

- Clearly define the novelty of this study compared to existing research.

- Provide a brief overview of similar interventions globally to strengthen the justification for the "pay-it-forward" approach.

Methods:

- Elaborate on the randomization process and how bias was minimized during participant allocation.

- Clarify whether participants were blinded to the intervention groups to reduce bias.

- Provide more detail on the operationalization of "intention to delay vaccination"—include thresholds or criteria.

Results:

- Expand the discussion of multivariate regression results to link findings back to public health implications.

- Clearly indicate whether variables included in the final model were pre-specified or selected post hoc based on significance.

Discussion:

- Deepen the analysis of cultural stigma and its implications for vaccination programs.

- Discuss global implications of the findings and their relevance beyond Iran.

- Critically evaluate the study’s limitations, particularly its reliance on a small, urban sample and absence of quantitative data.

- Suggest future research directions, such as mixed-method studies or larger cross-sectional analyses.

Conclusion:

- Expand on the limitations of the study, particularly concerning sample size and potential biases due to the socioeconomic distribution of participants.

- Discuss how the findings align or diverge from global HPV vaccination trends.

- Provide actionable recommendations for public health policymakers based on the findings.

Reviewer #2: This study, carried out in China with a relatively large N: Factor Influencing Caregivers' Action to Vaccinate Adolescent Girls Against HPV in China, is fairly well described. The method is replicable, and the process and outcome of the study are well understood. Nevertheless, several parts of the article are unclear and lack understanding for the reader.

my comments to make the article easier to read and understand:

Title: the study design should be added to the title.

Abstract: in the method section, explain how the data was collected.

Line 75 - 82: this section should go into the method section, to finish the background with the study objective.

Methods :

Even if the entire protocol has been published elsewhere (ref 16), certain information should be given in this article. In particular, the calculation of the sample size and the expected N of the study to see if the power of the study has been respected.

Line 119-122: this part should go in an ethics section at the end of the method.

Table 2: table 2 in its current format is very difficult to read, and it's hard to see which variable is linked to which OR. Statistically significant ORs should be highlighted more prominently. Since we have the 95% CIs, the statistical p's are not useful, which would save space and visibility in the table.

The discussion is complete and understandable, but I think it should be structured into sub-chapters, with a clearly defined and developed limit of the study.

6. PLOS authors have the option to publish the peer review history of their article (what does this mean? ). If published, this will include your full peer review and any attached files.

**Do you want your identity to be public for this peer review?** For information about this choice, including consent withdrawal, please see our Privacy Policy .

Reviewer #1: No

Reviewer #2: No

---

## [Author Response · Author response to Decision Letter 1]

16 Feb 2025

Journal Requirements:

We have thoroughly reviewed PLOS ONE’s style guidelines and ensured that our manuscript complies with all the formatting requirements.

As requested, we have amended the Role of Funder statement in the Funding section (line 396–397) to: “The funders had no role in study design, data collection and analysis, decision to publish, or preparation of the manuscript.” We have also updated the Funding Statement to: “The work reported in this publication is part of the research ‘Innovation Lab of Work Vaccine Delivery Research,’ supported by the Bill & Melinda Gates Foundation (INV-034554). The conclusions and opinions expressed in this work are those of the authors and shall not be attributed to the Foundation. Under the grant conditions of the Foundation, a Creative Commons Attribution 4.0 License has already been assigned to the Author Accepted Manuscript version that might arise from this submission.” We would appreciate your assistance in updating the online submission form accordingly.

In accordance with your request, the minimal de-identified data set necessary to replicate our study findings has been uploaded to a stable, public repository. The data can be accessed at [figshare] via the following link: [DOI:10.6084/m9.figshare.28424924]. We have also updated the Data Availability statement in the submission form accordingly.

Reviewers’ comments:

Reviewer #1: This study addresses a critical public health issue by investigating factors influencing HPV vaccination uptake among adolescent girls in China, a country facing significant cervical cancer burden. The findings provide valuable insights into caregiver decision-making and the impact of innovative interventions like the "pay-it-forward" model. However, methodological clarity, result interpretation, and presentation need improvement. I recommend a Major Revision.

Abstract

- Clarify the "pay-it-forward" intervention in a single concise sentence for accessibility to a broader audience.

RESPONSE We greatly appreciate this valuable comment. In response, we have added the definition of “pay-it-forward” in the Methods section of the abstract: “Pay-it-forward is a novel model that motivates participants in adopting healthy behaviors and making community contributions. In this study, it offers an individual a free shot of HPV vaccine and then asks whether they would like to donate to support another person to get the same vaccination.” (line 26-29).

Introduction

- Clearly define the novelty of this study compared to existing research.

RESPONSE: We appreciate the Reviewer’s valuable advice. We have clarified the novelty of this study in the background section. The revised sentence is:“This study addresses this gap by focusing on the actual decision-making process of caregivers, thus bridging the divide between willingness and action. In addition, the study introduces a pay-it-forward intervention as a novel approach to influence vaccination behavior, which has not been extensively studied in the context of HPV vaccination.”(line 81-84).

- Provide a brief overview of similar interventions globally to strengthen the justification for the "pay-it-forward" approach.

RESPONSE: Thank you for your insightful suggestion. To strengthen the justification for the "pay-it-forward" approach, we have expanded the background section to include examples of similar interventions globally. The revised text reads “Previous studies have demonstrated that the pay-it-forward approach effectively enhances health behavior engagement, including increasing screening rates for gonorrhea, chlamydia, and hepatitis among men who have sex with men, as well as improving influenza vaccination rates among children and the elderly. [16-18]” (line 87-90).

Methods:

- Elaborate on the randomization process and how bias was minimized during participant allocation.

RESPONSE: Thank you for your helpful comment. We have expanded on the randomization process in the Methods section. The revised text reads: “We obtained a de-identified contact list of adolescent girls aged 15–18 years from the residential committee through community health center staff. This list included all girls within this age range who have registered their names and contacts information at the community health center. We then performed simple random sampling to select eligible girls by generating a random number for each individual using Microsoft Excel. These individuals were sorted in ascending order based on their assigned random numbers. Recruitment proceeded by contacting individuals at the top of the list until the desired sample size was reached at each study site.” (line 112-119).

Additionally, we have elaborated on how bias was minimized during participant allocation. The revised text now states� “Upon arrival at the study site, participants were randomly assigned to either the standard-of-care arm or the pay-it-forward arm using sealed opaque envelopes to ensure allocation concealment. Each envelope contained a random numerical ID corresponding to the assigned arm (standard-of-care or pay-it-forward). The allocated intervention was concealed from both study participants and research assistants onsite prior to assignment until the envelope was opened, thereby maintaining blinding throughout the assignment process.” (line 120-126).

- Clarify whether participants were blinded to the intervention groups to reduce bias.

RESPONSE: Thank you for your insightful comment. In the revised manuscript, we have clarified that participants were blinded to the intervention groups in our study. The revised text states: “Upon arrival at the study site, participants were randomly assigned to either the standard-of-care arm or the pay-it-forward arm using sealed opaque envelopes to ensure allocation concealment. Each envelope contained a random numerical ID corresponding to the assigned arm (standard-of-care or pay-it-forward). The allocated intervention was concealed from both study participants and research assistants onsite prior to assignment until the envelope was opened, thereby maintaining blinding throughout the assignment process.” (line 120-126).

- Provide more detail on the operationalization of "intention to delay vaccination"—include thresholds or criteria.

RESPONSE: We appreciate the reviewer’s insightful comment. In the context of China’s limited supply and difficulty in securing appointments for 4v- and 9v- vaccines, we operationalized the “intention to delay vaccination” by including a specific question in the questionnaire. Caregivers were asked whether they planned to wait for the 4v- or 9v-HPV vaccines despite the availability of the domestically produced 2v-HPV vaccine. Caregivers who indicated their intent to wait for the 4v- or 9v- HPV vaccines were categorized as intending to delay vaccination.

This operationalization is now discussed in detail in the "Independent variables" section. Additionally, to provide further context, we have added the following statement: “In the context of China’s limited supply and difficulty in securing appointments for 4v- and 9v- HPV vaccines.” (line 189-190) to elaborate further.

Results:

- Expand the discussion of multivariate regression results to link findings back to public health implications.

RESPONSE: Thank you for your helpful suggestion. In the revised manuscript, we have expanded the discussion of the multivariate regression results and explicitly linked the findings to public health implications. Specifically, we have provided a more detailed interpretation of how factors such as caregiver occupation, knowledge of the HPV vaccine, and the intention to delay vaccination were associated with HPV vaccine uptake. These findings highlight critical areas where targeted health interventions can address both individual and systemic barriers to HPV vaccination.

Additionally, we have emphasized the relevance of our findings for regions with limited government-funded vaccination programs, such as parts of Africa and South Asia, where access to HPV vaccines is restricted. Our study underscores the importance of strategies such as financial subsidies and tailored health education, which could help improve HPV vaccination uptake in these regions and ultimately reduce the burden of cervical cancer. The revised text states: “HPV vaccination rates remain low in many countries, particularly in low-income regions where vaccine access is limited and healthcare systems may lack the infrastructure to promote the vaccine effectively[39]. The significant role of economic barriers and lack of knowledge about the vaccine mirrors challenges observed in other countries, such as India and several sub-Saharan African nations[40, 41]. Although this study was conducted in China, the findings are highly relevant to other regions facing similar barriers to HPV vaccination uptake. Many low-and middle-income countries, particularly those with limited government-sponsored vaccination programs, could benefit from strategies that combine financial support with community-driven health education initiatives. These approaches address both financial and informational barriers, which are prevalent globally.” (line 335-345).

- Clearly indicate whether variables included in the final model were pre-specified or selected post hoc based on significance.

RESPONSE Thank you for your valuable comment. The selection of variables for the final multivariable model was conducted post hoc and based on statistical significance after adjusting for caregivers' age and gender, as outlined in the statistical analysis section. Specifically, variables with a P-value less than 0.2 after adjustment were included in the model. This threshold was chosen to ensure that potentially important predictors were not excluded prematurely, while maintaining a balance between model complexity and interpretability. We have also emphasized this selection criterion in the results section, where it is stated: “Variables with a P-value less than 0.2 after adjusting for age and gender of caregivers were included in the multivariable model.” (line 246-247).

Discussion:

- Deepen the analysis of cultural stigma and its implications for vaccination programs.

RESPONSE: We appreciate the Reviewer’s comment and have expanded the analysis of cultural stigma in HPV vaccination within the discussion section. The revised text reads: “In addition to misinformation and vaccine availability, cultural stigma significantly impacts HPV vaccine uptake. In many societies, the HPV vaccine is often associated with sexual activity, leading to widespread misconceptions about its purpose and necessity. This stigma can deter caregivers from vaccinating adolescents due to concerns about societal judgment or misinterpretation of their intentions. To address this, public health campaigns should emphasize the vaccine’s role in preventing cervical cancer while incorporating culturally sensitive messaging while incorporating culturally sensitive messaging. These efforts can help to reduce stigma, promote acceptance, and improve vaccination uptake.” (line 325–334).

- Discuss global implications of the findings and their relevance beyond Iran.

RESPONSE: We appreciate the reviewer’s insightful comment. While this study was conducted in China, we have taken the suggestion into account and expanded the discussion to address the broader global implications of our findings. The following text has been added: “Although this study was conducted in China, the findings are highly relevant to other regions facing similar barriers to HPV vaccination uptake. Many low-and middle-income countries, particularly those with limited government-sponsored vaccination programs, could benefit from strategies that combine financial support with community-driven health education initiatives. These approaches address both financial and informational barriers, which are prevalent globally. Future research could explore how the“pay-it-forward” model might be adapted to improve HPV vaccination rates in diverse settings, tailoring the intervention to align with local context, cultural nuances, and available resources.” (line 339–348).

- Critically evaluate the study’s limitations, particularly its reliance on a small, urban sample and absence of quantitative data.

RESPONSE We appreciate the reviewer’s valuable suggestion. In response, we have revised the limitations section to provide a more critical evaluation of the study’s sample and data limitations. The original sentence has been updated from: “As a study embedded within a randomized controlled trial, the sample size was limited and not specifically powered to assess all variables influencing vaccination behavior” to “As a study embedded within a randomized controlled trial, the sample size was relatively small and limited to an urban sample, which may restrict the generalizability of the findings to rural areas or other regions with different socio-cultural contexts.” (line 350-353). Our study is indeed a quantitative analysis. However, we understand that the reviewer’s concern may stem from the limited range of variables considered in the study. While our current model focused on the most significant factors (e.g., caregiver age, gender, and vaccine preferences), we acknowledge that the study could benefit from a more extensive set of quantitative variables, such as detailed socio-economic status, health literacy, and psychological factors, which could provide a more nuanced understanding of vaccination behavior.

- Suggest future research directions, such as mixed-method studies or larger cross-sectional analyses.

RESPONSE We appreciate the reviewer’s valuable suggestion. We have revised the conclusion section to address your suggestion. The following text has been added: “Future research should aim to address these limitations by conducting larger, more diverse cross-sectional studies to improve the representativeness of findings. Mixed-methods studies combining quantitative and qualitative approaches could also provide deeper insights into the cultural, socioeconomic, and psychological barriers to HPV vaccination.” (line 377–381).

Conclusion:

- Expand on the limitations of the study, particularly concerning sample size and potential biases due to the socioeconomic distribution of participants.

RESPONSE: Thank you for your valuable feedback. We have revised the conclusion section to address your suggestion. The following text has been added: “However, there are several limitations in this study. The relatively small and urban-focused sample size may restrict the generalizability of our findings to rural or socioeconomically diverse populations. Additionally, potential biases may arise from the socioeconomic distribution of participants, which could influence the observed associations.” (line 373–377).

- Discuss how the findings align or diverge from global HPV vaccination trends.

RESPONSE: Thank you for your insightful comment. In response to your suggestion, we have expanded our discussion to better contextualize our findings within global HPV vaccination trends. Specifically, we have highlighted that “HPV vaccination rates remain low in many countries, particularly in low-income regions where vaccine access is limited and healthcare systems may lack the infrastructure to promote the vaccine effectively. The significant role of economic barriers and lack of knowledge about the vaccine mirrors challenges observed in other countries, such as India and several sub-Saharan African nations.” (line 339-333). Additionally, we have included a discussion on how the findings diverge from global HPV vaccination trends, noting that “Caregivers who intended to delay HPV vaccination were less likely to vaccinate their adolescent daughters, suggesting that the intention to delay vaccination may be a particular and specific barrier in China.” (line 306-308).

- Provide actionable recommendations for public health policymakers based on the findings.

RESPONSE: Thank you for your valuable feedback. We have revised the conclusion to make the recommendations more specific and actionable for public health policymakers. The updated version emphasizes targeted health education for underserved communities, financial subsidies for low-income families, and community involvement through

---

## [Decision Letter · Decision Letter 1]

PONE-D-24-51311R1Factors Influencing Caregivers’ Action to Vaccinate Adolescent Girls Against HPV:  Secondary Analysis of a Randomized Controlled Trial in ChinaPLOS ONE

Dear Dr. Li,

Thank you for submitting your manuscript to PLOS ONE. After careful consideration, we feel that it has merit but does not fully meet PLOS ONE’s publication criteria as it currently stands. Therefore, we invite you to submit a revised version of the manuscript that addresses the points raised during the review process.

We look forward to receiving your revised manuscript.

Kind regards,

Omar Enzo Santangelo

Academic Editor

PLOS ONE

**Additional Editor Comments:**

Dear Authors, the manuscript needs major revisions, please respond point by point to the reviewers' requests.

Kind regards

Reviewers' comments:

Reviewer's Responses to Questions

**Comments to the Author**

1. If the authors have adequately addressed your comments raised in a previous round of review and you feel that this manuscript is now acceptable for publication, you may indicate that here to bypass the “Comments to the Author” section, enter your conflict of interest statement in the “Confidential to Editor” section, and submit your "Accept" recommendation.

Reviewer #3: All comments have been addressed

Reviewer #4: (No Response)

2. Is the manuscript technically sound, and do the data support the conclusions?

Reviewer #3: Yes

Reviewer #4: Partly

3. Has the statistical analysis been performed appropriately and rigorously? 

Reviewer #3: Yes

Reviewer #4: Yes

4. Have the authors made all data underlying the findings in their manuscript fully available?

Reviewer #3: Yes

Reviewer #4: Yes

5. Is the manuscript presented in an intelligible fashion and written in standard English?

Reviewer #3: Yes

Reviewer #4: Yes

6. Review Comments to the Author

Reviewer #3: (No Response)

Reviewer #4: This manuscript focuses on investigating factors influencing caregivers' decisions to vaccinate adolescent girls against HPV in China through a secondary analysis of a randomized controlled trial. The study has addressed some important public health issues and has been revised based on reviewers' comments, but there is still room for improvement in multiple aspects. Here are detailed revision suggestions:

Title and Abstract

Title: The addition of "Secondary Analysis of a Randomized Controlled Trial in China" in the title clarifies the study design, which is a good improvement. However, it could be made more concise. Consider rephrasing to something like "Factors Affecting Caregivers' HPV Vaccination Decisions for Adolescent Girls: A Chinese RCT Secondary Analysis" to enhance readability.

Abstract: While the definition of the "pay - it - forward" intervention and data collection method have been added, it could be further improved. For example, briefly mention the key results more quantitatively. Instead of just stating the factors associated with vaccination uptake, provide approximate percentages or odds ratios in a more summary - oriented way to give readers a quicker understanding of the magnitude of the effects.

Introduction

Literature Review: Although the background provides an overview of the HPV vaccination situation in China and the importance of the study, it could be more comprehensive. Cite more recent and diverse literature, especially studies that have explored the gap between willingness and action in HPV vaccination in different cultural or geographical contexts. This would strengthen the justification for the current study.

Research Gap: The statement of the research gap is clear, but it could be more specific. Explain precisely how the "pay - it - forward" intervention fills the gap in the existing literature, not just in terms of influencing vaccination behavior but also in comparison to other similar interventions that may have been tried in the HPV vaccination field.

Methods

Study Design and Participants: The randomization process and blinding description have been improved. However, it would be beneficial to provide more details about the characteristics of the four community health service centers in Chengdu. For example, mention their location (urban/rural distribution, socioeconomic characteristics of the surrounding areas), as this could potentially influence the results and the generalizability of the findings.

Sample Size Calculation: The addition of sample size calculation details is a great improvement. To further enhance transparency, explain why a 10% non - response or dropout rate was assumed. Provide some references or prior data to support this assumption.

Variable Measurements: For the "intention to delay vaccination" variable, while the operationalization is now clear, consider validating this measure. You could mention if there were any pilot studies or pre - testing of this question to ensure that it accurately captures the concept. Also, for other variables, such as "vaccine hesitancy," describe in more detail how the responses were coded and categorized.

Statistical Analysis: Clearly state the assumptions of the logistic regression models. For example, discuss the assumption of independence of observations and how it was met in the context of this study. Also, explain why a backward stepwise strategy was chosen over other variable selection methods, and discuss its potential limitations.

Results

Presentation of Results: The results section is generally well - presented. However, in Table 1, provide more information about the missing data. Report the number of missing values for each variable, and if possible, discuss how missing data were handled. In Table 2, consider adding a column to show the direction of the association (e.g., positive or negative) for each variable, especially for readers who may quickly scan the table.

Interpretation of Results: Expand on the interpretation of the results in the text. For example, when discussing the significant factors, compare the magnitudes of the odds ratios in more detail. Explain why the effect of the "pay - it - forward" intervention on vaccination uptake is of a certain magnitude, and how it compares to other factors in practical terms.

Discussion

Comparison with Other Studies: The discussion of cultural stigma has been expanded, which is good. However, when comparing the study findings with other regions, provide more in - depth analysis. For example, compare the prevalence of cultural stigma - related barriers in China with those in other Asian countries or globally. Analyze how cultural differences may lead to different levels of stigma and its impact on vaccination rates.

Global Implications: The global implications section could be more focused. Instead of a broad statement about low - and middle - income countries, identify specific regions or countries where the "pay - it - forward" model could be most applicable. Provide some preliminary ideas on how the model might need to be adapted for different cultural, economic, and healthcare system contexts.

Limitations: The limitations section has been improved, but it could be more comprehensive. Discuss the potential impact of the COVID - 19 guidelines on the intervention in more detail. For example, how did the new guidelines specifically affect participants' willingness to visit clinics? Were there any differences in the impact across different groups of participants? Also, consider adding a limitation related to the self - reported nature of the data, as there may be recall bias or social desirability bias in caregivers' responses.

Future Research Directions: The suggestions for future research are reasonable. To make them more actionable, provide some initial steps or potential research questions for the proposed mixed - methods studies or larger cross - sectional analyses. For example, in a mixed - methods study, what specific qualitative data could be collected to complement the quantitative findings?

Conclusion

Summary of Findings: The conclusion should more clearly summarize the key findings in a concise manner. List the most important factors influencing HPV vaccine uptake in a bullet - point format or a short, numbered list for easy reference.

Recommendations: The recommendations have been made more specific, but they could be further operationalized. For example, when suggesting financial subsidies for low - income families, provide some estimates of the amount of subsidy needed, or how it could be implemented in different regions of China. Also, describe in more detail how local health workers can be involved in fostering community engagement, such as the types of training they may need.

Overall Structure and Clarity

Consistency: Check for consistency in the use of terms throughout the manuscript. For example, ensure that "caregivers" and "parents" are used consistently, and if there are differences, clarify them.

Readability: Some sentences are quite long and complex, especially in the discussion section. Break them into shorter, more digestible sentences to improve readability. Use more subheadings in the discussion section to further organize the content, even if not strictly in a sub - chapter format. This would help readers quickly locate and understand different aspects of the discussion.

7. PLOS authors have the option to publish the peer review history of their article (what does this mean? ). If published, this will include your full peer review and any attached files.

**Do you want your identity to be public for this peer review?** For information about this choice, including consent withdrawal, please see our Privacy Policy .

Reviewer #3: **Yes: ** Awoke Derbie

Reviewer #4: No

---

## [Author Response · Author response to Decision Letter 2]

5 Mar 2025

Reviewers’ comments:

Reviewer #4: This manuscript focuses on investigating factors influencing caregivers' decisions to vaccinate adolescent girls against HPV in China through a secondary analysis of a randomized controlled trial. The study has addressed some important public health issues and has been revised based on reviewers' comments, but there is still room for improvement in multiple aspects. Here are detailed revision suggestions:

Title and Abstract

Title: The addition of "Secondary Analysis of a Randomized Controlled Trial in China" in the title clarifies the study design, which is a good improvement. However, it could be made more concise. Consider rephrasing to something like "Factors Affecting Caregivers' HPV Vaccination Decisions for Adolescent Girls: A Chinese RCT Secondary Analysis" to enhance readability.

RESPONSE: Thank you for your insightful feedback on the title. We agree that a more concise title would improve readability. In response, we have revised the original title from “Factors Influencing Caregivers’ Action to Vaccinate Adolescent Girls Against HPV: Secondary Analysis of a Randomized Controlled Trial in China” to “Factors Affecting Caregivers’ HPV Vaccination Decisions for Adolescent Girls: A Secondary Analysis of a Chinese RCT.”

Abstract: While the definition of the "pay - it - forward" intervention and data collection method have been added, it could be further improved. For example, briefly mention the key results more quantitatively. Instead of just stating the factors associated with vaccination uptake, provide approximate percentages or odds ratios in a more summary - oriented way to give readers a quicker understanding of the magnitude of the effects.

RESPONSE: We appreciate the reviewer’s valuable suggestion and have made the recommended adjustments to the abstract for greater clarity. Specifically, we have incorporated a more concise summary of the key results with quantitative details. The revised sentence is as follows:“Daughters of caregivers who were previously unaware of the HPV vaccine were three times more likely to be vaccinated (OR=3.01, 95% CI: 1.27-7.14). Caregivers who did not intend to delay vaccination had daughters with eight times higher vaccination uptake (OR=8.26, 95% CI: 4.36-15.67). Participation in the ‘pay-it-forward’ intervention increased vaccination rates by more than twofold (OR=2.22, 95% CI: 1.19-4.15). Daughters of unemployed or retired caregivers had nearly four times higher vaccination rates compared to those whose caregivers were employed (OR=3.97, 95% CI: 1.81-8.72). Prior refusal of the vaccine by caregivers was associated with an 80% reduction in vaccination uptake among daughters (OR=0.23, 95% CI: 0.06-0.81).” (lines 40-49)

Introduction

Literature Review: Although the background provides an overview of the HPV vaccination situation in China and the importance of the study, it could be more comprehensive. Cite more recent and diverse literature, especially studies that have explored the gap between willingness and action in HPV vaccination in different cultural or geographical contexts. This would strengthen the justification for the current study.

RESPONSE: Thank you for your insightful feedback on the literature review section. We appreciate your suggestion to provide a more comprehensive overview and to include a broader range of recent, diverse literature, particularly studies that explore the gap between willingness and action in HPV vaccination across different cultural and geographical contexts. In response, we have revised the literature review to incorporate additional recent studies, expanding the scope to include research from various regions and populations that address this gap. This inclusion helps strengthen the justification for our study by contextualizing the findings in a broader, global framework.

The following text has been added to enhance the discussion:“Recent studies from regions such as India, Southeast Asia, and Europe have highlighted similar gaps between vaccine willingness and actual uptake[16-18]. These findings underscore the importance of understanding not only the factors that influence willingness to vaccinate, but also the broader socio-cultural, economic, and healthcare system factors that play a critical role in shaping actual vaccine uptake.” (lines 79-84)

Research Gap: The statement of the research gap is clear, but it could be more specific. Explain precisely how the "pay - it - forward" intervention fills the gap in the existing literature, not just in terms of influencing vaccination behavior but also in comparison to other similar interventions that may have been tried in the HPV vaccination field.

RESPONSE: Thank you for your constructive feedback. In response to your suggestion, we have revised the manuscript to more explicitly describe how the "pay-it-forward" intervention addresses gaps in the existing literature. The updated explanation clarifies the unique contribution of this intervention compared to other strategies in the HPV vaccination field.

The original sentence was revised from:“In the 'pay-it-forward' model, an individual receives a gift (e.g., a subsidized HPV vaccine) and is then asked whether they would like to donate to support another person. Previous studies have demonstrated that the pay-it-forward approach effectively enhances health behavior engagement, including increasing screening rates for gonorrhea, chlamydia, and hepatitis among men who have sex with men, as well as improving influenza vaccination rates among children and the elderly[16-18].”to:“While various interventions, such as health education and financial incentives, have been explored to increase HPV vaccination uptake, they have often faced limitations in overcoming socio-cultural barriers and promoting sustained behavior change[19, 20]. In contrast, the “pay-it-forward” model not only provides a tangible benefit (e.g., a subsidized HPV vaccine) but also leverages social reciprocity by encouraging individuals to donate or support others. This unique mechanism taps into intrinsic social motivations, which has shown effectiveness in improving engagement in other health behaviors, such as increasing STI screening rates among men who have sex with men and flu vaccination rates among children and elderly populations [21-23]. The novelty of this intervention lies in its potential to bridge the gap between willingness and actual vaccine uptake by aligning social norms and altruistic behavior with public health goals.” (lines 91-103)

Methods

Study Design and Participants: The randomization process and blinding description have been improved. However, it would be beneficial to provide more details about the characteristics of the four community health service centers in Chengdu. For example, mention their location (urban/rural distribution, socioeconomic characteristics of the surrounding areas), as this could potentially influence the results and the generalizability of the findings.

RESPONSE: We sincerely thank the reviewer for this constructive suggestion. To provide greater context on the characteristics of the four community health service centers in Chengdu and their potential influence on the results and generalizability of our findings, we have made the following revisions to the "Study Design and Participants" section: “The four sites were selected to reflect varying socio-economic contexts based on the relative level of individual disposable income in 2021. This approach ensures a broad representation of different socio-economic environments within Chengdu. The selected sites are as follows: Site A: Located in the most developed urban area, this site benefits from higher healthcare access and represents a higher socio-economic status with better public health infrastructure. Site B: Situated in a higher middle-income suburban area, this site experiences relatively good healthcare access but is influenced by suburban socio-economic factors. Site C: This site represents a lower middle-income suburban area with more limited healthcare resources and socio-economic challenges. Site D: The least developed area in terms of infrastructure, healthcare access, and socio-economic conditions. This site faces the greatest challenges related to economic factors and healthcare delivery.” (lines 122-133)

Sample Size Calculation: The addition of sample size calculation details is a great improvement. To further enhance transparency, explain why a 10% non - response or dropout rate was assumed. Provide some references or prior data to support this assumption.

RESPONSE: We sincerely thank the reviewer for this helpful suggestion. We have provided additional details regarding the basis for the 10% non-response or dropout rate assumption to enhance transparency. The 10% dropout rate was primarily informed by the findings from the pre-test conducted in our pilot study in Chengdu. In this pilot phase, we observed a certain level of participant attrition, with approximately 10% of participants not completing the study. Given the context of our study, where participants may experience logistical or personal challenges in completing the trial, this pilot data provided a conservative estimate for the dropout rate. While we did not find direct literature references for a 10% dropout rate specific to similar studies, the pilot data provided a reasonable estimate to ensure adequate statistical power.

We have now incorporated this additional explanation in the revised manuscript. The original sentence has been updated from: “Considering a 10% non-response or dropout rate, 133 participants per arm are needed.” to: “Considering a 10% non-response or dropout rate based on results from the pilot study, 133 participants per arm are needed.” (line 172).

Variable Measurements: For the "intention to delay vaccination" variable, while the operationalization is now clear, consider validating this measure. You could mention if there were any pilot studies or pre - testing of this question to ensure that it accurately captures the concept. Also, for other variables, such as "vaccine hesitancy," describe in more detail how the responses were coded and categorized.

RESPONSE: Thank you very much for your insightful comments regarding the operationalization and validation of the "intention to delay vaccination" and "vaccine hesitancy" variables. We appreciate your suggestions and have revised the manuscript to address them in more detail.

Intention to Delay Vaccination:

To ensure that the "intention to delay vaccination" variable accurately captures the intended concept, we conducted a pilot study prior to the main trial. In this pilot study, we pre-tested the survey instrument with a small, diverse sample of participants from the target population to assess the clarity, comprehensibility, and relevance of the questions. Based on the feedback received, adjustments were made to enhance the clarity and specificity of the wording. We have added the following text to the manuscript to clarify this process:“To ensure that this measure accurately captures the concept of intention to delay vaccination, we conducted a pilot study in which the wording of this question was pre-tested for clarity and understanding with a small group of caregivers from the study’s target population.” (lines 216-219).

Vaccine Hesitancy:

Vaccine hesitancy was assessed by retrospectively examining whether participants had ever experienced hesitation, postponement, or refusal of HPV vaccination. Specifically, participants were asked the following questions:“Have you ever postponed the HPV vaccination (except for allergies)?”“Have you ever refused to receive the HPV vaccination (except for allergies)?”“Have you ever heard negative information about HPV vaccination?” These questions were designed to capture past instances of vaccine-related hesitation and decision-making rather than to determine an overall hesitancy score. Each response was recorded separately to provide insights into different aspects of past vaccine-related behaviors. This approach ensures a clear and detailed understanding of participants' prior experiences with HPV vaccination. We have updated the manuscript to include this clarification: “Vaccine hesitancy was assessed by asking participants to indicate their level of agreement with three statements regarding past behavior and attitudes: “Have you ever been hesitant about getting the HPV vaccination (except for allergies)?”; “Have you ever postponed the HPV vaccination (except for allergies)?”; and “Have you ever refused to receive the HPV vaccination (except for allergies)?” Each response recorded separately to reflect different aspects of past vaccine-related decision-making.” (lines 228-234).

Statistical Analysis: Clearly state the assumptions of the logistic regression models. For example, discuss the assumption of independence of observations and how it was met in the context of this study. Also, explain why a backward stepwise strategy was chosen over other variable selection methods, and discuss its potential limitations.

RESPONSE: Thank you for your insightful comment regarding the assumptions of the logistic regression models and the rationale behind the selection of the backward stepwise method. In response to your suggestion, we have revised the manuscript to provide a more detailed explanation of the assumptions underlying the logistic regression models, as well as a clearer justification for the choice of the backward stepwise strategy.

Assumptions of Logistic Regression Models:

We appreciate your suggestion to elaborate on the assumptions made when applying logistic regression models. The following text has been added to the manuscript to clearly outline the assumptions:“The logistic regression models in this study assumed the independence of observations, linearity of the logit, and the absence of multicollinearity. To ensure the independence of observations, participants were randomized at the individual level, and no repeated measures were taken from the same individual.” (line 249-253).

Rationale for Backward Stepwise Strategy and Potential Limitations:

The backward stepwise method was chosen for its systematic approach to model selection, in which non-significant variables are iteratively removed to achieve a more parsimonious and interpretable model. This strategy helps identify the most important predictors and eliminates irrelevant variables, thereby improving model simplicity and interpretability. We have added the following text to justify this choice:“This method was chosen for its systematic elimination of non-significant variables and ability to simplify the model. However, it carries risks of overfitting, exclusion of important interactions, and dependence on the initial variable set.” (lines 246-249).

Results

Presentation of Results: The results section is generally well - presented. However, in Table 1, provide more information about the missing data. Report the number of missing values for each variable, and if possible, discuss how missing data were handled. In Table 2, consider adding a column to show the direction of the association (e.g., positive or negative) for each variable, especially for readers who may quickly scan the table.

RESPONSE: Thank you for your valuable feedback on the presentation of the results. In response to your suggestions, we have made the following revisions to improve transparency and readability:

1. Missing Data in Table 1: We appreciate your suggestion to provide more detailed information about missing data in Table 1. In the revised manuscript, we have included the number of missing values for the "age" variable. Specifically, there were 19 missing values for this variable, which accounts for a very small proportion of the total sample. We did not observe missing data for any other variables in the dataset, and therefore, no imputation or special handling was required. We have updated the table’s note to clearly indicate this information:“There were 19 missing values for the variable ‘age’.” (line 278).

2. Direction of Association in Table 2: In response to your suggestion to include a column indicating the direction of the association in Table 2, we have added a new column titled "Direction of Association." This column shows

---

## [Decision Letter · Decision Letter 2]

PONE-D-24-51311R2Factors Affecting Caregivers’ HPV Vaccination Decisions for Adolescent Girls: A Secondary Analysis of a Chinese RCTPLOS ONE

Dear Dr. Li,

Thank you for submitting your manuscript to PLOS ONE. After careful consideration, we feel that it has merit but does not fully meet PLOS ONE’s publication criteria as it currently stands. Therefore, we invite you to submit a revised version of the manuscript that addresses the points raised during the review process.

**ACADEMIC EDITOR: **

Dear Authors, the manuscript needs minor revisions, please respond point by point to the reviewers' requests.

Kind regards

We look forward to receiving your revised manuscript.

Kind regards,

Omar Enzo Santangelo

Academic Editor

PLOS ONE

Journal Requirements:

Additional Editor Comments :

Dear Authors, the manuscript needs minor revisions, please respond point by point to the reviewers' requests.

Kind regards

Reviewers' comments:

Reviewer's Responses to Questions

**Comments to the Author**

1. If the authors have adequately addressed your comments raised in a previous round of review and you feel that this manuscript is now acceptable for publication, you may indicate that here to bypass the “Comments to the Author” section, enter your conflict of interest statement in the “Confidential to Editor” section, and submit your "Accept" recommendation.

Reviewer #5: (No Response)

Reviewer #6: All comments have been addressed

2. Is the manuscript technically sound, and do the data support the conclusions?

Reviewer #5: Yes

Reviewer #6: Yes

3. Has the statistical analysis been performed appropriately and rigorously? 

Reviewer #5: Yes

Reviewer #6: Yes

4. Have the authors made all data underlying the findings in their manuscript fully available?

Reviewer #5: Yes

Reviewer #6: Yes

5. Is the manuscript presented in an intelligible fashion and written in standard English?

Reviewer #5: Yes

Reviewer #6: Yes

6. Review Comments to the Author

Reviewer #5: Dear authors,

Thank you for the manuscript. I can see a great deal of work has gone into the writing and revising of the manuscript. I want to congratulate you on your work and contribution to this topic. I believe the manuscript could benefit from some minor revisions before publication:

1. Methods: The subsidy amount is mentioned, but not the total cost of the vaccine. This is necessary context for the reader.

2. Results: The percentage of caregivers who paid it forward is not mentioned in the results. My assumption, from seeing the pilot publication, is that this result will be shared in the main RCT publication. Please reference where this can be found, as it leaves the reader wondering.

3. Results: Am I correct to assume that “Have you ever heard of HPV”, “Annual family income”, and “Is there anyone around who has been vaccinated against HPV” were not <0.2 after adjusting for age and gender as they were not included in the multivariable analysis? Please mention why these were not included in the multivariable analysis.

4. Results & Discussion: If available, please provide the number of participants approached in screening, how many were sent the questionnaire, and how many were finally included for this analysis. This is vital to understand potential bias in the results and contextualise findings, such as a higher vaccine uptake in the non-intervention group being attributed to active engagement. In addition, if high drop-out rates were found, please reflect on their impact on the results in the limitations.

I wish you a smooth further publication of the manuscript and look forward to seeing the published article.

Kind regards

Reviewer #6: This manuscript presents a well-conceived and timely analysis exploring the factors that influence caregivers’ decisions to vaccinate adolescent girls against HPV in China. The use of data from a RCT, combined with a behavioral model (Andersen’s Health Service Utilization Model), gives the findings both scientific and practical weight. The authors have responded comprehensively to prior reviewer comments, significantly improving the clarity, depth, and utility of the manuscript. Overall, the study contributes valuable evidence to the growing literature on HPV vaccine uptake in low- and middle-income settings, particularly in culturally complex environments such as China.

Line 217: “were more likely to vaccinate their adolescents” → consider rephrasing to "were more likely to have their daughters vaccinated" for smoother flow.

Line 294: “ensuring access to vaccination services for participants willing to consider other formulations” could be made more concise.

7. PLOS authors have the option to publish the peer review history of their article (what does this mean? ). If published, this will include your full peer review and any attached files.

**Do you want your identity to be public for this peer review?** For information about this choice, including consent withdrawal, please see our Privacy Policy .

Reviewer #5: No

Reviewer #6: No

---

## [Author Response · Author response to Decision Letter 3]

21 Apr 2025

Reviewer #5:

1. Methods: The subsidy amount is mentioned, but not the total cost of the vaccine. This is necessary context for the reader.

RESPONSE: Thank you for this valuable suggestion. To address your comment, we have added the following information to the manuscript: :“The total cost for a full three-dose HPV vaccine series in the local market ranged from approximately CNY 990 to CNY 3,900 (USD 139–546), depending on the type and manufacturer.” (Lines 160-162)

2. Results: The percentage of caregivers who paid it forward is not mentioned in the results. My assumption, from seeing the pilot publication, is that this result will be shared in the main RCT publication. Please reference where this can be found, as it leaves the reader wondering.

RESPONSE: We appreciate your thoughtful observation. The proportion of caregivers who paid it forward will indeed be detailed in the main RCT manuscript, which is currently under peer review. As these data are not yet publicly available, we are unable to cite them at this time. We will ensure to update the reference in future versions. Thank you for your understanding.

3. Results: Am I correct to assume that “Have you ever heard of HPV”, “Annual family income”, and “Is there anyone around who has been vaccinated against HPV” were not <0.2 after adjusting for age and gender as they were not included in the multivariable analysis? Please mention why these were not included in the multivariable analysis.

RESPONSE: Thank you for this important clarification request. The variables “Have you ever heard of HPV,” “Annual family income,” and “Is there anyone around who has been vaccinated against HPV” had p-values less than 0.2 after adjusting for age and gender, and therefore, were included in the multivariable model. However, they were later excluded through backward stepwise logistic regression, which retained only the most significant predictors of HPV vaccine uptake. This has been clarified in the manuscript (Lines 312–313).

4. Results & Discussion: If available, please provide the number of participants approached in screening, how many were sent the questionnaire, and how many were finally included for this analysis. This is vital to understand potential bias in the results and contextualise findings, such as a higher vaccine uptake in the non-intervention group being attributed to active engagement. In addition, if high drop-out rates were found, please reflect on their impact on the results in the limitations.

RESPONSE: Thank you for highlighting this important point. A total of 321 participants were approached during screening and all 321 were sent the questionnaire. The questionnaire completion rate was 100% in both the pay-it-forward and standard-of-care arms. Thus, all participants were included in the final analysis. This information has been added to the manuscript (Lines 267–269). As no participant dropped out or declined to respond, we believe potential bias due to non-response or attrition is minimal.

Reviewer #6:

Line 217: “were more likely to vaccinate their adolescents” → consider rephrasing to "were more likely to have their daughters vaccinated" for smoother flow.

RESPONSE: Thank you for this helpful stylistic suggestion. We have revised the phrase accordingly to “were more likely to have their daughters vaccinated” at both Line 286 and Line 300 in the manuscript for improved clarity and flow.

Line 294: “ensuring access to vaccination services for participants willing to consider other formulations” could be made more concise.

RESPONSE: Thank you for pointing this out. We have revised the phrase to a more concise version: "ensuring access to vaccination services for participants willing to consider other formulations" to make it more concise. The revised version is now as follows: " ensuring access to vaccination for participants considering other formulations"(Lines 434-435)

---

## [Editor Report · Decision Letter 3]

Factors Affecting Caregivers’ HPV Vaccination Decisions for Adolescent Girls: A Secondary Analysis of a Chinese RCT

PONE-D-24-51311R3

Dear Dr. Li,

We’re pleased to inform you that your manuscript has been judged scientifically suitable for publication and will be formally accepted for publication once it meets all outstanding technical requirements.

Kind regards,

Omar Enzo Santangelo

Academic Editor

PLOS ONE
---

## [Editor Report · Acceptance letter]

PONE-D-24-51311R3

PLOS ONE

Dear Dr. Li,

I'm pleased to inform you that your manuscript has been deemed suitable for publication in PLOS ONE. Congratulations! Your manuscript is now being handed over to our production team.

Kind regards,

on behalf of

Dr. Omar Enzo Santangelo

Academic Editor

PLOS ONE